

# Exploring nonlinear associations between atmospheric new-particle formation and ambient variables: an information theoretic approach

Martha A. Zaidan[1,2,3], Ville Haapasilta[3], Rishi Relan[4], Pauli Paasonen[1], Veli-Matti Kerminen[1], Heikki Junninen[1,5], Markku Kulmala[1,6], and Adam S. Foster[3,7,8]

[1]Institute for Atmospheric and Earth System Research/Physics, Helsinki University, FI-00560, Helsinki, Finland
[2]Aalto Science Institute, School of Science, Aalto University, FI-00076, Espoo, Finland
[3]Dept. of Applied Physics, Aalto University, FI-00076, Espoo, Finland
[4]Dept. of Applied Mathematics and Computer Science, Technical University of Denmark, 2800 Kongens Lyngby, Denmark
[5]Institute of Physics, University of Tartu, Ülikooli 18, EE-50090 Tartu, Estonia
[6]Aerosol and Haze laboratory, Beijing University of Chemical Technology, 100096 Beijing, China
[7]WPI Nano Life Science Institute (WPI-NanoLSI), Kanazawa University , Kakuma-machi, Kanazawa 920-1192, Japan
[8]Graduate School Materials Science in Mainz, Staudinger Weg 9, 55128, Germany

*Correspondence to:* Martha A Zaidan (martha.zaidan@helsinki.fi), Pauli Paasonen (pauli.paasonen@helsinki.fi, Veli-Matti Kerminen (veli-matti.kerminen@helsinki.fi) and Markku Kulmala (markku.kulmala@helsinki.fi)

**Abstract.** Atmospheric new particle formation (NPF) is a very non-linear process that includes atmospheric chemistry of precursors and clustering physics as well as subsequent growth before NPF can be observed. Thanks to ongoing efforts, now there exists a tremendous amount of atmospheric data, obtained through continuous measurements directly from the atmosphere. This fact makes the analysis by human brains difficult, but on the other hand enables to usage of modern data science techniques. Here, we calculate and explore *the mutual information* between observed NPF events (measured at Hyytiälä, Finland) and a wide variety of simultaneously monitored ambient variables: trace gas and aerosol particle concentrations, meteorology, radiation and a few derived quantities. The purpose of the investigations is to identify key factors contributing to the NPF. The applied mutual information method finds that the formation events correlate with sulfuric acid concentration and water content, ultraviolet radiation, condensation sink and temperature. Previously, these quantities have been well-established to be important players in the phenomenon via dedicated field, laboratory and theoretical research. The novelty of this work is to demonstrate that the same results are now obtained by a data analysis method which operates without supervision and physical insight. This suggests that the method is suitable to be implemented widely in the atmospheric field to discover other interesting phenomena and its relevant variables.

## 1 Introduction

New particle formation (NPF) is an important source of aerosol particles and cloud condensation nuclei (CCN) and in a vast number of atmospheric environments ranging from remote continental areas to heavily-polluted urban centers (Kulmala et al., 2004; Dunne et al., 2016; Wang et al., 2017). The occurrence and strength of NPF and its influence on the CCN budget in different atmospheric environments depends on a delicate balance between the factors that favor NPF and subsequent particle



growth and the factors that suppress these processes (Kerminen and Kulmala, 2002; Pierce and Adams, 2007; Westervelt et al., 2014; Kulmala et al., 2017) . As a result, researchers have not managed to find a general framework, or formulae, on how to relate atmospheric NPF to the concentrations of various trace gases, meteorological quantities and radiation parameters.

Based on data from field measurements, several studies investigated the relations between NPF and meteorological condi-

tions (Nilsson et al., 2001) and various chemical compounds (Bonn and Moortgat, 2003; Kulmala et al., 2004; Almeida et al., 2013; Nieminen et al., 2014). Such studies have found the ideal conditions for NPF events to consist of low atmospheric water content, low preexisting particle concentration and high solar radiation (Boy and Kulmala, 2002). In addition, sulfuric acid is believed to be the single most important compound to participate in the atmospheric NPF (Kerminen et al., 2010; Sipilä et al., 2010; Petäjä et al., 2011; Nieminen et al., 2014).

Due to the practical limitations, the measurement campaigns typically last from weeks to months and they often have a dedicated focus. On one hand such approach enables a very detailed inspection for somewhat narrower scope, but on the other hand there is a risk of overlooking important processes falling outside the chosen, predetermined scope. One way to circumvent this issue is to have long-term continuous measurements of a wide variety of atmospheric variables. Nowadays there is more and more focus on continuous observations as described by Kulmala (2018). However, such enterprises then open a new set

problems: how to analyse all the collected data? It is clear that techniques offered by the modern data science, such as data mining and machine learning, should be consulted.

Previously, Hyvönen et al. (2005) carried out a comprehensive study using two different data mining methods on eight years of continuous measurements of eighty variables from Hyytiälä, Finland. Their first method was based on unsupervised K-means clustering. Here, the first step was to obtain features, by calculating the mean and standard deviation of selected

atmospheric variables. The calculated features were then compressed using a principal component analysis to reduce the high data dimensions. The first six of the obtained principal components, containing most of data information, were fed into K-means clustering. The method found that the data sets can be separated into four clusters. The first two clusters represent NPF event days whereas the other two clusters reflect non-event days. The centers of the clusters could be traced back to the atmospheric variables. The method demonstrated that relative humidity, global radiation and sensible heat have data separation power and

correlate with NPF. In addition to those, their results indicated that ozone ($O_3$) and carbon dioxide ($CO_2$) concentrations might also correlate with NPF.

The second method used by Hyvönen et al. (2005) was based on supervised machine learning classification. Several machine learning models (such as linear discriminant analysis, support vector machine, logistic regression) were set up to perform a classification task for each day as an event or a non-event day. The goal was not to separate event days from non-event days,

but in understanding which atmospheric variables should be used to clearly separate the two groups. In this case, the mean and standard deviation of atmospheric variables were calculated as the input whereas the aerosol particle formation event and non-event days database was used as the output. Due to the initial model's random parameters, the models were run 1000 times using different training and test sets to ensure the result stability. The selected models used a pair and triplet combination of atmospheric variables. The models were ranked based on the classification performance and the best model was used to

evaluate all pair and triplet combinations of the atmospheric variables. Finally, the supervised classification models found that



the best pair of atmospheric variables to classify events/non-events are condensation sink and relative humidity. The latter was also found through clustering method.

The results of Hyvönen et al. (2005) support some earlier conclusions from Boy and Kulmala (2002) stating that NPF events are largely explained by three parameters: temperature, water content and radiation. However, they did not find significant correlations between NPF and radiation variables. Although the classification methods might be a suitable tool for finding correlation between variables in complex systems, the used implementation is not effective for this case. The first reason concerns the used features, such as mean and standard deviations. This pratice compresses the measurement data into a single quantity for each day, that may potentially lead to information loss in the data. Secondly, the implementation procedure is computationally expensive. This requires the exploration of all possible models and variable combinations to find the best pairs. The models also need to be run multiple times to ensure their stability.

To overcome the above-mentioned issues, we propose here an alternative method - based on information theory - to be used in atmospheric data analysis. *Mutual information* (MI), one of the many information quantities, measures the amount of information that can be obtained about one random variable by observing another one. In this paper, MI is first introduced and then used to find the maximal amount of shared information between atmospheric variables and NPF. In other words, the goal is to find the most relevant atmospheric variables in relation to NPF events using a data-driven information theoretic method based on the dataset measured at Hyytiälä, Finland.

## 2 Atmospheric database measured at SMEAR II station Hyytiälä, Finland

In this study, we utilize the data measured during the years 1996-2014 at Station for Measuring Forest Ecosystem-Atmosphere Relations (SMEAR) II station in Hyytiälä, Finland, operated by Helsinki University (SMEAR website, 2017).

### 2.1 Sampling site

The SMEAR II station is located in Hyytiälä forestry field station in southern Finland (61° 51′N, 24° 17′E, 181 m above sea level), about 220 km northwest of Helsinki. It also lies between two large cities, Tampere and Jyväskylä, that are about 60 km and 90 km from the measurement site, respectively. Homogeneous 55-year-old (in 2017) scots-pine-dominated forests surround the station. SMEAR II is classified as a rural background site considering the levels of air pollutants, shown by e.g. submicron aerosol number size distributions (Asmi et al., 2011a; Nieminen et al., 2014).

The SMEAR II station has been established for multidisciplinary research, including atmospheric sciences, soil chemistry and forest ecology. The station consists of a measurement building, a 72-meter high mast, a 15-meter tall tower and two mini-watersheds. It is equipped with extensive research facilities for measurement of various gases concentration, various fluxes, meteorological parameters (e.g. temperature, wind speed and direction, relative humidity) solar and terrestrial radiation (e.g. ultraviolet rays) and atmospheric aerosols (e.g. particle size distribution). The measurements for forest ecophysiology and productivity, such as photochemical reflectance, and the measurements for soil and water balance, also take place there. A





detailed description of the continuous measurements performed at this station can be found in Kulmala et al. (2001a); Hari and Kulmala (2005); SMEAR website (2017).

## 2.2  Measured variables

In this study, we used four types of continuous measurement data: gas concentrations, meteorological conditions, radiation
variables and aerosol particle concentrations. The gases include nitrogen monoxide (NO) and other oxides ($NO_x$), ozone ($O_3$), sulfur dioxide ($SO_2$), water ($H_2O$), carbon dioxide ($CO_2$) and carbon monoxide (CO). Meteorological data include the temperature, humidity, pressure, and wind speed and direction, among others. The gas concentrations and meteorological data measurements are performed at the heights of 4.2, 8.4, 16.8, 33.6, 50.4 and 67.2 m. The radiation variables include UV-A, UV-B, PAR, global, net, reflected global and reflected PAR. These measurements are mostly performed at radiation tower (18
m). The aerosol particle number size distributions (dry diameter of 3-1000 nm) are measured at 35 m height.

Table 1 collects all the atmospheric variables used in this study, including the adapted shorthand notation used throughout the current paper together with few details on the measurements. The raw data can be accessed free of charge via the SMEAR website (2017), which also contains more information on the measurements. It should be noted that not all the measured atmospheric variables are included in the current analysis.

## 15  2.3  Derived variables

In addition to directly measured variables, there are few derived variables included in this study. Aerosol particle condensation sink (CS) determines how rapidly molecules and small particles condense onto preexisting aerosol particles and it is strongly related the shape of the size distribution (Pirjola et al., 1999; Kulmala et al., 2001b). CS is formulated as:

$$CS = 4\pi D \sum_i \beta_{M_i} r_i N_i \tag{1}$$

where $r_i$ is the radius of a particle for size class $i$, $N_i$ is the particle concentration in the respective class $i$, $D$ is the diffusion coefficient of the condensing vapor and $\beta_M$ is the transitional correction factor, defined in Fuks and Sutugin (1970).

Sulfuric acid ($H_2SO_4$) concentration is included in the study since it is believed to be one of key factors in atmospheric aerosol particle formation (Nieminen et al., 2014). Unfortunately, there are no continuous long-term measurements of sulfuric acid concentrations at SMEAR II Hyytiälä. In order to gauge sulfuric acid, we need to calculate its proxy concentration based
on the measured gas concentrations, solar radiation and the measured aerosol particle size distributions acting as CS (Kulmala et al., 2001b). Weber et al. (1997) proposed a direct proxy calculation utilizing hydroxyl (OH) radical concentration, given by:

$$p_1 = k_1 . \frac{[SO_2].[OH]}{CS} \tag{2}$$

where $[SO_2]$ and $[OH]$ are the concentrations of sulfur dioxide and hydroxyl, respectively. CS is condensation sink and the
parameter $k_1$ is the constant number of $2.2 \times 10^{-12} cm^3 molec^{-1} s^{-1}$. In our case, this proxy cannot be calculated because there is no continuous measurement of hydroxyl (OH) concentration at SMEAR II Hyytiälä. Fortunately, Petäjä et al. (2009)





proposed other two proxies by using CS and solar radiation in UV-B range as well as global radiation (Glob). The proxy formulations are given by:

$$p_2 = k_2 . \frac{[SO_2].UV\text{-}B}{CS} \tag{3}$$

$$p_3 = k_3 . \frac{[SO_2].Glob}{CS} \tag{4}$$

where $k_2$ and $k_3$ are median values for the scaling factors, that are $9.9 \times 10^{-7} \text{m}^2\text{W}^{-1}\text{s}$ and $2.3 \times 10^{-9} \text{m}^2\text{W}^{-1}\text{s}$, respectively. Here, we include the proxies 2 and 3 ($p_2$ and $p_3$) calculated for the years 1996-2014 in our analysis.

Finally, it is essential to have a database of aerosol particle formation days – without such database the correlation analysis between NPF and atmospheric variables cannot be performed. We used a database of the years 1996-2014, generated by the atmospheric scientists at Helsinki University. The database has been created by visual inspection of the continuously measured aerosol size distributions over a size range of 3-1000 nm at SMEAR II Hyytiälä forest (Dal Maso et al., 2005). The method classifies days into three main groups: event, non-event and undefined days. An event day occurs when there is a growing new mode in the nucleation size range prevailing over several hours, whilst a non-event day takes place when the day is clear of all traces of particle formation. Finally, an undefined day is assumed when it cannot be unambiguously classified as either an event or non-event day. For simplicity, in the current study we do not consider the undefined days and those are removed from our database. Figure 1 shows two examples of the day when NPF and growth (event day) and the day when no particle formation is observed (non-event day) on April 2005 at Hyytiälä station. The x-axis displays the 24-hours time period whilst y-axis denotes the range of particle diameters (from 3 nm to 1000 nm). The color indicates the particle concentration level (cm$^{-3}$).

## 3   Computational Methods: concept and their application

Before the raw data can be fed into an analysis model, they need to be pre-processed first and these steps will be outlined below. After that, the Mutual Information (MI) method will be introduced.

### 3.1   Data pre-processing

The (raw) data used in this paper ranges from 1 January 1996 to 31 December 2014, totalling 18 years. The first step in pre-processing is to exclude the undefined days, as the focus is to find the correlation between aerosol particle formation days and atmospheric variables. In order to reduce the amount of irrelevant data, we then eliminate night-time data points in all atmospheric variables. When the atmospheric photo-chemistry is most intense (during the daytime), the strongest and long-lasting events of the atmospheric NPF are typically observed (Kulmala and Kerminen, 2008; Nieminen et al., 2014). Due to significant variation in daytime and night-time in Hyytiälä forest during a year, it is necessary to use accurate sunrise and sunset times (Duffett-Smith and Zwart, 2011; National Oceanic and Atmospheric Administration, 2017). Finally, to ensure that all the variables have an equal weight, the data need to be normalized to have zero mean and unit variance. Otherwise, variables with large numerical values may appear more important in the analysis.





## 3.2 Information Theory: A brief introduction

Information theory is a mathematical representation of the conditions and parameters affecting the transmission and processing of information (Stone, 2015). Information theory has been applied to a wide range of applications, such as communication (Xie and Kumar, 2004), cryptography (Bruen and Forcinito, 2011) and seismic exploration (Mukerji et al., 2001). This subsection introduces briefly the basic concepts of information quantities, as well as the definitions and notations of probabilities that will be used throughout the paper. In-depth explanation concerning the principles of information theory can be found for example in MacKay (2003); Cover and Thomas (2012); Stone (2015).

### 3.2.1 Entropy

Entropy is a key measure in information theory. It quantifies the amount of uncertainty involved in the value of a random variable. If $\mathbb{X}$ is the set of all data points $\{x_1, \cdots, x_N\}$ that $X$ could take, and $p(x)$ is the probability of some $x \in \mathbb{X}$, then the entropy of $X$, $H(X)$, is defined as

$$H(X) \equiv -\sum_{x \in \mathbb{X}} p(x) \log p(x). \tag{5}$$

Using the concept of information entropy $H(X)$, one can further define two related and useful quantities: the joint and conditional entropies.

*Joint entropy* measures the amount of uncertainty in two random variables $X$ and $Y$ taken together, and it is defined by

$$H(X,Y) \equiv -\sum_{x \in \mathbb{X}, y \in \mathbb{Y}} p(x,y) \log p(x,y), \tag{6}$$

where the random variable $Y$ can take values from the set of points $\mathbb{Y} = \{y_1, \cdots, y_N\}$ and $p(x,y)$ is the joint probability of $x$ and $y$.

*Conditional entropy* quantifies the amount of uncertainty remaining in the random variable $Y$ when the value of random variable $X$ is known. This can be defined mathematically by

$$H(Y|X) \equiv -\sum_{x \in \mathbb{X}} p(x) \sum_{y \in \mathbb{Y}} p(y|x) \log p(y|x) = -\sum_{x \in \mathbb{X}, y \in \mathbb{Y}} p(x,y) \log \frac{p(x,y)}{p(x)}, \tag{7}$$

where $p(y|x)$ is the conditional probability of $y$ given $x$ satisfying the chain rule of probability: $p(x,y) = p(y|x)p(x)$. It follows directly from the defintion (7) that conditional entropy fulfils the property

$$H(Y|X) = H(X,Y) - H(X), \tag{8}$$

which relates the two-variable conditional and joint entropies with the single-variable information entropy.

### 3.2.2 Mutual Information

The mutual information (MI) of two random variables is a measure of the mutual dependence between these two variables. MI is thus a method for measuring the degree of relatedness between data sets. MI and its relation to joint and conditional entropies



is illustrated visually in Figure 2 with the help of correlated variables $X$ and $Y$. The left disk (red and orange surface area) shows the entropy $H(X)$, while the right disk (yellow and orange surface area) shows the entropy $H(Y)$. The total surface area covered by the two disks is the joint entropy $H(X,Y)$. The conditional entropy $H(X|Y)$ is the red surface on the left, while the conditional entropy Y given X, $H(Y|X)$, is the yellow surface area on the right. The intersection of the red and yellow disks, the orange surface area in the middle, is the mutual information $I(X;Y)$ between $X$ and $Y$.

More formally the mutual information of $X$ relative to $Y$ is given as

$$I(X;Y) \equiv H(X,Y) - H(X|Y) - H(Y|X) = H(X) + H(Y) - H(X,Y). \tag{9}$$

From the equation (9) it is clear that MI is symmetric with respect to the variables $X$ and $Y$. In terms of probabilities, MI is given by:

$$I(X;Y) \equiv \sum_{x \in \mathbb{X}, y \in \mathbb{Y}} p(x,y) \log \frac{p(x,y)}{p(x)\,p(y)}. \tag{10}$$

From the defintion (10), one can see that for completely independent and uncorrelated variables, $p(x,y) = p(x)p(y)$, the MI vanishes, as expected. It can be also seen that in the other extreme where the variables are the same, MI reduces into the corresponding information entropy.

MI has found its use in modern science and technology, for example in search engines (Su et al., 2006), in bioinformatics (Lachmann et al., 2016), in medical imaging (Cassidy et al., 2015) and in feature selection (Peng et al., 2005). Probably at least a part of MI method's appeal comes from its capability to effectively measure non-linear correlation between data sets (Steuer et al., 2002; Chen et al., 2010). In this aspect MI is superior to the standard Pearson correlation coefficient (PCC) (Pearson, 1895), which is only suitable for measuring linear correlation (Wang et al., 2015). To illustrate this, Figure 3 shows a comparison between PCC (commonly represented by $\rho$) and MI using a standard test set of linearly and non-linearly correlated data. The upper row shows six linear data sets, whereas the bottom row plots six non-linear data sets; both rows also contain one uncorrelated data set (the middle one). Both methods estimate similar correlation for the linear data sets and correctly detect the uncorrelated data. In the case of non-linear data PCC simply fails, whereas MI method is able to measure the correlation in the data.

MI implementation is straightforward for discrete distributions because the required probabilities for calculating MI can be computed precisely based on counting. However, MI implementation for continuous distributions may be tricky because the probability distribution function is often unknown. A binning method can be implemented for calculating MI involving continuous distribution. This method makes the data completely discrete by grouping the data points into bins in the continuous variables. Nevertheless, the choice of binning size (i.e. the number of data points per bin) is a non-trivial task, since this choice often leads to different MI result. The binning method does not allow MI calculation between two data sets that have different resolution - this would be a major obstacle in this study. Therefore, in the current investigation we will use the so-called nearest neighbour method (Kraskov et al., 2004; Ross, 2014). It has been shown to be accurate, insensitive to the choice of model parameter and also computationally relatively fast.





### 3.3 Mutual Information implementation: nearest neighbor method

This subsection explains the nearest neighbor MI method. Suppose $x$ is a discrete variable and $y$ is a continuous variable. The method computes a number $I_i$ for each data point $i$, based on its nearest-neighbors in the continuous variable $y$. First, using Euclidean distance (or other types of distance metrics), we find the $k$-th closest neighbor to point $i$ among $N_{x_i}$, where $N_{x_i}$ is

the data points whose value of the discrete variable equals $x_i$. This results in $d$, that is the distance to this $k$-th neighbor. Next, we count the number of neighbors $m_i$ in the full data set that lie within distance $d$ to point $i$ (including the $k$-th neighbor itself). Based on $N_{x_i}$, and $m_i$, MI for every data point $i$ can be computed using:

$$I_i = \psi(N) - \psi(N_{x_i}) + \psi(k) - \psi(m_i) \tag{11}$$

where $N$ is the number of full data points and $k$ is the user choice for the number of nearest neighbor. The symbol $\psi(.)$ is the

digamma function, defined as the logarithmic derivative of the gamma function. This can be expressed as:

$$\psi(z) = \frac{d}{dz}\ln\big(\Gamma(z)\big) = \frac{\Gamma'(z)}{\Gamma(z)} \tag{12}$$

where $\Gamma(.)$ is a gamma function. The detailed explanation about gamma and digamma functions can be found in Abramowitz and Stegun (2012).

After obtaining MI for every point $i$, in order to estimate the MI from our data set, we average $I_i$ over all data points,

symbolized by $\langle . \rangle$, to give:

$$I(X;Y) = \langle I_i \rangle \tag{13}$$
$$= \psi(N) - \langle\psi(N_x)\rangle + \psi(k) - \langle\psi(m)\rangle \tag{14}$$

where $k$ is determined by a user. In order to bound the MI estimates within the interval $(-1, 1)$ and make it comparable with Pearson correlation coefficient (PCC) (Pearson, 1895), the proposed scaling factor from Numata et al. (2008) is used, to give:

$$\hat{I}(X;Y) = \text{sign}\big[I(X;Y)\big]\sqrt{1 - \exp(-2|I(X;Y)|)} \tag{15}$$

where sign is a signum function and $|.|$ is the absolute value. In this case, the negative values of $\hat{I}(X;Y)$ should not be interpreted as anti-correlations.

Figure 4 illustrates the concept of nearest neighbor MI method. This MI implementation is capable for analyzing two data sets with different time resolutions. This motivates the adoption of the method in this study, where the time resolution between the

measured atmospheric variables and the classification of aerosol particle formation days is not uniform. Hence, the calculation of time-domain features, such as the mean and standard deviation, is not required here. These features naturally compress the data and typically lead to information loss. The subfigure (a) illustrates the time-series measurement of an atmospheric variable for each day. Every single day can be associated to two classes that are event (E) or non-event days (NE). It can be seen that there are multiple measurements in a day whereas there is only single event/non-event data available for each day.

The distances between the measurement vectors themselves are then calculated as illustrated in the subfigure (b). Here, we





take the example of day index 100 ($D_{100}$). Here, the distance between the measurement vectors at $D_{100}$ from the same class is calculated. In the subfigure (c), the distance vector of $D_{100}$ calculated from the same class (event days) is then ranked in ascending order, shown on the top line. In this particular case, the user choice parameter $k$-th closest neighbor is selected to be 3. So the distance threshold is found at $D_{98}$. The distance vector from the same class (the red sign) is then projected on the

bottom line. The bottom line contains the distance vector of $D_{100}$ calculated from all classes. The dashed line, representing the threshold from point distance $D_{100}$ out to the 3-rd neighbor is drawn until the bottom line. After that, it is found that the number of distance points which is the 3-rd closest neighbor to $D_{100}$ on the top lines is the 7-th closest neighbor on the bottom line ($m = 7$). The above processes point out that the parameter $m$ becomes a crucial factor in MI estimator, shown in the equation 14. This parameter is obtained through the above processes involving the distances calculation between different

data resolution. This is advantageous in computing MI between event classification data and atmospheric variable data, that typically vary in different time resolution. In summary, besides its effectiveness in estimating non-linear correlation, the nearest neighbor MI is also advantageous for the current problem because 1) it is a non-parametric method making no assumptions about the functional form (Gaussian or non-Gaussian) of the statistical distribution underlying the data, 2) there is no need for computationally costly binning to generate histograms, 3) it is computationally fairly light and 4) the model contains only one

free model parameter ($k$) and it is easy to tune.

Prior to demonstrate the result of MI application on the atmospheric data in section 4, the following subsection discusses first how MI is capable in estimating non-linear relationship, tested on a simulated physics equation.

### 3.4 Mutual information: a simulation case study

MI capability in detecting a non-linear relationship between two variables on an artificial benchmark dataset is already illus-

trated in Figure 3. Before applying nearest-neighbour MI to real atmospheric data, this subsection shows another, more physical case study demonstrating how well MI is able to detect non-linear relationship between two correlated variables.

We consider the intensity of blackbody radiation. The monochromatic emissive power of a blackbody $F_B(\lambda)[\mathrm{Wm}^{-2}\mu\mathrm{m}^{-1}]$ is related to temperature $T$ and wavelength $\lambda$ by (Seinfeld and Pandis, 2016):

$$F_B(\lambda) = \frac{2\pi c^2 h \lambda^{-5}}{e^{ch/k\lambda T} - 1} \tag{16}$$

where $k$ is Boltzmann constant ($k = 1.381 \times 10^{-23} \mathrm{JK}^{-1}$), $h$ is a Planck constant ($6.626 \times 10^{-34}\mathrm{Js}$) and $c$ is the speed of light in vacuum ($c = 2.9979 \times 10^8 \mathrm{m/s}$). The solar spectral irradiance at the top of the Earth's atmosphere at 5777 K is shown in Figure 5a. If the temperature is varied (randomly between 10 K to 10000 K in this case), the solar spectral irradiance for the same range of wavelengths looks quite different, as is shown in Figure 5b. The correlation level for both scenarios using PCC (again symbolised by $\rho$) and the nearest-neighbour MI is also shown. It can be seen that when the temperature is fixed,

Pearson correlation is still able to detect the correlation between wavelength and solar spectral irradiance, but fails detecting the relationship between these variables when the data is more messy due to the variation in the temperature. On the other hand, MI is able to detect the correlation between $\lambda$ and $F_B(\lambda)$ in both cases.



## 4 Results

The result section is divided into two subsections. The first part presents the result of MI correlation analysis between atmospheric variables and NPF. The second part then discusses the scatter plot of several relevant atmospheric variables to NPF.

### 4.1 Correlation analysis between atmospheric variables and NPF

In this study, the atmospheric variables are continuous values while the aerosol formation days classification is discrete. Hence, we implemented the MI based on nearest neighbor method for finding the correlation between these two data sets, explained earlier in subsection 3.3. Figure 6 presents the correlation results in the form of bar charts, including gases and aerosols (top), meteorology (middle) and radiation (bottom). Several atmospheric variables are measured at different heights, such as gas concentrations and meteorological parameters. In this case, the mean and standard deviation of their MI correlation level were

calculated. For those variables, the rectangular bar represents the mean of MI correlation level whereas the whisker is its two standard deviations. For the variables which are measured only at one particular height/location, their MI correlation is only represented as the rectangular bar without any whisker.

The top subplot in Figure 6 shows the MI correlation level between NPF and gas concentrations as well as aerosol (CS). It can be seen that the water concentration ($H_2O$) has the highest correlation among others. This finding is in agreement with

those presented by Boy and Kulmala (2002) and Hyvönen et al. (2005). The reason for the high MI correlation between NPF occurrence and $H_2O$ concentration has so far not been explained. Whether this relation is truly causal or appears because of correlations in diurnal or annual cycles of air masses related to other NPF related variables remains to be assessed in future studies. The second highest correlation variable in this group is condensation sink (CS). The high correlation with CS can be expected, since CS describes the main sink for vapours participating in NPF and it is also an effective sink for freshly formed

new particles. Previous studies have shown that the average value of CS is typically lower on NPF days compared with non-event days (Dal Maso et al., 2007; Asmi et al., 2011b; Dada et al., 2017). Furthermore, this subplot shows that sulfuric acid ($H_2SO_4$), evaluated using two proxies, correlates well with NPF. It is known that $H_2SO_4$ is one of the key vapours participating in NPF (Kulmala et al., 2013). The correlation between NPF and $H_2SO_4$ has been proven through analysis on the data obtained from a number of measurement sites (Kuang et al., 2008; Nieminen et al., 2009; Paasonen et al., 2010; Wang et al., 2011) as

well as in laboratory experiments (Almeida et al., 2013).

The MI found that ozone ($O_3$) and carbon dioxide ($CO_2$) might be related to the NPF process. The correlation of these variables were also indicated by Hyvönen et al. (2005) via a K-means clustering method. The correlation with $O_3$ is probably related to the formation of extremely low volatile organic compounds (ELVOC), which can be initiated by the ozonolysis of monoterpenes (Ehn et al., 2014). ELVOCs are presumed to participate in NPF. The correlation with $CO_2$, on the other hand,

might be related to the coupling between photosynthesis and emission of monoterpenes, as suggested by Kulmala et al. (2014).

On the other hand, the result suggests that sulfur dioxide ($SO_2$) and nitrogen oxides ($NO_x$) do not correlate strongly with NPF. The $SO_2$ observation is inconclusive: its concentration has been found to be higher for NPF event days in some studies (Boy et al., 2008; Young et al., 2013) and lower in others (Wu et al., 2007; Dai et al., 2017). Previously, Boy and Kulmala



(2002) already stated that in the cases of $SO_2$ and $NO_x$ at this measurement site, there are no significant differences found between event and non-event days.

The middle subplot presents the MI correlation level for all measured meteorological variables. Some variables with the subscript *ave*, are averages of meteorological variables measured at different heights. As the top subplot, we calculated the mean

and standard deviation of their MI correlation level and display them as a rectangular bar with a whisker. The middle subplot shows that there is a very strong correlation between NPF and relative humidity ($RHURAS_{ave}$ as well as RHTd). A similar result was also reported in Hyvönen et al. (2005). On NPF event days, the average ambient RH is typically lower than non-event days in both clean and polluted environments (Vehkamäki et al., 2004; Hamed et al., 2007; Jun et al., 2014; Qi et al., 2015; Dada et al., 2017). High values of RH tend to have a negative influence on the solar radiation intensity, photochemical reactions and

atmospheric lifetime of aerosol precursor vapors (Hamed et al., 2011). Our result points out that the temperature ($T_{ave}$ and $T_d$) correlates with NPF, as also observed by Boy and Kulmala (2002) and Hyvönen et al. (2005) for this site. The correlation with temperature might be related emissions of monoterpenes, which is a strong function of temperature (Guenther et al., 1995). However, the temperature is associated with so many atmospheric variables (e.g. boundary layer height, turbulence, radiation, RH, and the volatility of the vapours) that the correlation might be caused by several different variables

In contrast, wind speed ($WS_{ave}$ and $WSU_{ave}$) and wind direction ($WD_{ave}$ and $WDU_{ave}$) have little correlation with NPF. Similar results were also reported by Boy and Kulmala (2002). They stated that the small correlation persists due to pollution from the westsouth - west (station building and city of Tampere). The correlation between NPF and rain indicator (SWS) as well as the atmospheric pressure (Pamb0) at Hyytiälä were also found to be weak. Several other meteorological variables (not displayed) were excluded from the analysis due to the data scarcity. It is also important to note that on both subplots (top and

middle), the whiskers for most bar variables are very short. This means that the MI correlation level for the same variables measured at various heights is similar. The whisker for wind speed ($WSU_{ave}$) is slighly longer because the measured wind speed varies moderately at different heights.

The bottom subplot shows the MI level of several radiation variables. It can be seen that most radiation variables have a strong relation with NPF. This fact was discussed earlier by Boy and Kulmala (2002), especially on the variable Ultraviolet A

($UV_A$). The high level of correlation in the global radiation (Glob) was also found by Hyvönen et al. (2005). In all measurement sites, the average solar radiation intensity tend to be higher on NPF event days compared with non-event days (Birmili and Wiedensohler, 2000; Vehkamäki et al., 2004; Hamed et al., 2007; Kristensson et al., 2008; Pierce et al., 2014; Qi et al., 2015; Wonaschütz et al., 2015). Radiation is known as the driving force for atmsopheric chemistry, producing low volatility vapours (e.g. sulfuric acid, ELVOCs) that participate in NPF.

The correlation between concentrations of particles with different sizes from 3 to 1000 nm and NPF is illustrated as a colored panel in Figure 7. There are four columns in the x-axis. The first three columns represent three periods between years 1996-2014 where each period comprises the correlation level for 6 years. The last column is the total correlation level for 18 years. The period division observes if the correlation level for all periods is similar and consistent. The y-axis shows the aerosol particle sizes. There are 51 ranges of particles size in the x-axis of the colored panel, but we *down-sample* the 51 particles size

ranges to be only 11 sizes for simplification. The color bar represents the MI correlation level between the specified aerosol




particles and NPF. It can be seen that NPF correlates very well with particles in the nucleation mode size range (3-25 nm). This can be expected, since in a relatively clean environment, such as Hyytiälä, NPF is the main source of nucleation mode particles. Clear correlations between particle concentrations and NPF event occurrence is also detected in the size range from 150 to 550 nm. In this size range, the correlation can be expected, since it is the concentration of these particles that has the

largest impact on the condensation sink (CS). Thus, the high concentrations of 150-550 nm particles disfavour NPF and the correlation can be presumed to be negative (see the text related CS in the top panel of Figure 6).

## 4.2 Scatter plot analysis

In order to understand in depth the results from the aforementioned MI analysis, a scatter plot matrix was generated, as shown in Figure 8. The plot involves some of the most important atmospheric variables in the NPF process, according to via the

MI analysis made in the previous subsection, including the sulfuric acid concentration ($H_2SO_4$), average temperature ($T_{ave}$), relative humidity ($RHT_d$), global radiation (Global) and condensation sink (CS). The logarithm was applied to the variables $H_2SO_4$ and CS to ease the scatter plot visualization. The red and blue dots in the plot represent event and non-event days, respectively. Along the diagonal are histogram plots of each column of x. Here, the same data as the above study was used (e.g. 18 years SMEAR II data sets). The undefined days were excluded. Next, the daily mean of all measurements during the day

time was computed and then normalized (between 0 and 1). Finally, for MI comparison, we performed a linear correlation to analyze the relationship between atmospheric variables and event/non-event days. Since the latter is a dichotomous variable (i.e. it contains two categories or discrete), we used Point-biserial correlation coefficient ($r_{pb}$), which is mathematically equivalent to PCC (Howell, 2012). This correlation coefficient is displayed on each histogram.

First, we focus on the histogram plots located on the sub-axes along the diagonal. It can be seen that the event and non-event

days are well separated in the cases of Global and $RHT_D$. These histogram plots demonstrate very well that NPF has a positive ($r_{pb} = 0.639$) and negative ($r_{pb} = -0.707$) correlation with the variables Global and $RHT_D$, respectively. When the value of Global radiation is high, NPF days are likely to occur. On the other hand, non-event days tend to take place when RH is high. The correlation between NPF and these variables were found earlier by MI in the previous subsection. This fact supports the view that the MI method is an effective tool to provide early correlation detection between atmospheric variables and NPF.

The next focus is on the variables $H_2SO_4$, CS and $T_{ave}$. The variable $H_2SO_4$ can still be detected through the linear correlation method ($r_{pb} = 0.403$). The histogram plot of $H_2SO_4$ shows that NPF event days do not take place when the concentration of $H_2SO_4$ is very low, whereas the event days usually occur when it is high. However, both event and non-event days may take place if the $H_2SO_4$ concentration level is medium (i.e. see the intersection between the red and blue histograms). Nevertheless, the scatter plots between Global, $RHT_D$ and $H_2SO_4$ indicate that these variables are connected to result in NPF. It is known

that the formation of 3 nm particles occurs on the days with strong solar radiation. In other words, to form $H_2SO_4$ in the atmosphere, high solar radiation is typically required. Likewise, high $H_2SO_4$ concentration in the atmosphere increase cluster formation and growth rate and hence favor the occurence of an NPF event (Almeida et al., 2013; Kulmala et al., 2013). On the other hand, when $RHT_D$ value is high, the radiation is typically low and therefore the $H_2SO_4$ concentration also tends to be low.





The above conclusion would be very challenging to be made by using a linear correlation analysis for variables $T_{ave}$ ($r_{pb} = 0.134$) and CS ($r_{pb} = 0.007$). These correlation coefficients do not reveal that the variables are related to NPF, which we know from previous literature results. Likewise, by observing the histogram plots, both event and non-event days may take place on any values of $T_{ave}$, except in very low or very high temperature regimes. This situation is also similar for the case of CS

for which event and non-event days are not separable. Since the histogram plots of CS and $T_{ave}$ present the complication in understanding their connection with NPF, their scatter plots should also be analyzed. For instance, the event and non-event days seem to be separated on the scatter plots between $T_{ave}$ and Global as well as $RHT_D$, where the last two variables are known to be correlated with NPF. This may explain how they are connected, but their correlation may be non-linear, since the separation takes place in the middle of the plot. Likewise, the scatter plots between CS and the variables Global, $RHT_D$ as well as $H_2SO_4$

show a separation between the event and non-event days. Even though the separation is not perfect, this may still clarify how they are connected.

This subsection demonstrates the analysis complexity by observing the histogram and scatter plots for some variables, such as $H_2SO_4$, $T_{ave}$ and CS. The intricacy might occur because the relationship among some atmospheric variables and NPF may be complex, non-linear or indirect, in addition to which there might be other variables influencing the process of NPF. CS

and temperature are known to impact NPF directly or indirectly, as dicussed in section 4.1. A sole investigation through linear correlation analysis, histogram and scatter plots for finding the relationship among atmospheric variables sometimes poses a challenge. This problem explain why MI should be used in the first place for finding early correlation detection between atmospheric variables as well as its phenomena.

## 5  Conclusions

This paper extends and complements the analysis of a previous data mining study on atmospheric data, conducted by Hyvönen et al. (2005). Both papers exploit the strengths of data-driven methods, but there are two notable distinctions between this study and the previous investigation. First, our work utilizes 18 years (1996-2014) of atmospheric measurements from the SMEAR II station in Hyytiälä, Finland. This means that the current work deals with ten more years of data. The utilization of a larger data set is expected to provide more reliable results and thus more accurate conclusion. Second, instead of using

data mining methods based on clustering and classification, this paper promotes the use of MI for identifying the key variables in atmospheric aerosol particle formation. The applied nearest-neighbour MI method is a powerful and computationally light tool capable of finding both linear and non-linear relationships between the measured atmospheric variables and observed NPF events. The method also contains only one free parameter (the number of nearest neighbours $k$) and its value does not affect the results significantly. Furthermore, the method operates directly on the data and does not require the calculation of

characterising compressed features (i.e. mean, standard deviation etc.) which might potentially lead to a partial information loss.

The MI method reports very similar findings with the previous atmospheric studies. The water content and sulfuric acid concentration are found to be strongly correlated with NPF. Furthermore, the results also suggest that NPF is influenced by





temperature, relative humidity, CS and radiation. According to the results from the MI analysis, the measurements taken at different heights have similar correlation with NPF.

As shown in the previous subsection, this method is more powerful than a linear correlation analysis. Therefore, this method should be used in the first place before performing a deeper data analysis method, such as through histogram and scatter plots.

5 This method could act as an early correlation detection for any atmospheric variables.

This work uses the longest available data sets of NPF observations with simultaneously measured ambient variables. As future works, we will seek to investigate the use of the method on different atmospheric data sets. For instance, robust correlation analysis is required for understanding other variables influencing atmospheric process, such as volatile organic compounds (VOC) and aerosol particles at sizes below 3 nm.

10 In order to enrich the analysis, the database from other SMEAR stations as well as previous research campaigns should be included. The data may contain more variation because they are measured in different locations. One anticipated obstacle is the scarceness of NPF days classification databases. Although an automatic classification algorithm to create such a database has been called for (Kulmala et al., 2012), currently the event/non-event days are laboriously classified using a manual visualization method (Dal Maso et al., 2005). There has been an attempt to use machine learning for automating aerosol database

15 classification, but the performance has not been completely satisfactory yet (Zaidan et al., 2017). One possibility to enhance the performance of the machine learning classification is to use the correlated atmospheric variables found in this study as an additional inputs for such models. Similar concept can also be applied in developing any atmospheric process or proxy. A proxy dependent variables can be selected by finding the most correlated variables to the interested proxy via MI.



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

**Figures**

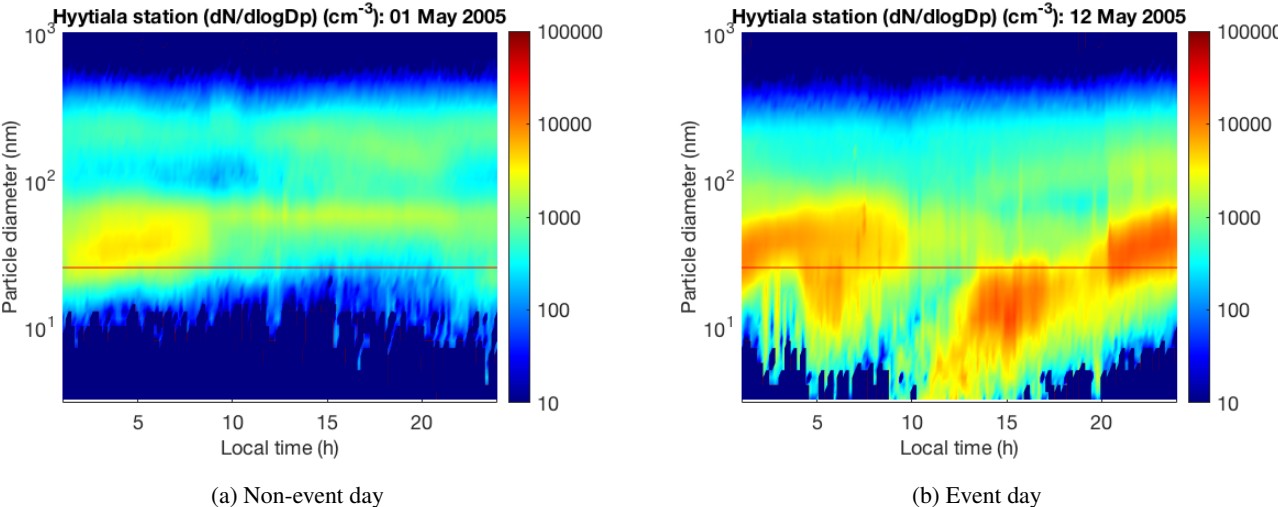

(a) Non-event day          (b) Event day

**Figure 1.** Examples of non-event and event days at Hyytiälä, Finland, in May 2005. A non-event day (a) is assumed when the day is clear of
all traces of particle formation whilst an event day (b) occurs when there is a growing new mode in the nucleation size range prevailing over
several hours. Data accessed via Smart-SMEAR (Junninen et al., 2009).



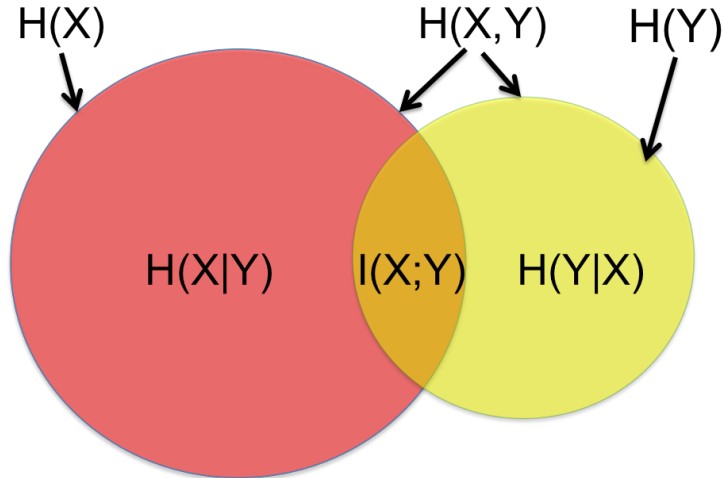

**Figure 2.** Venn diagram of entropy properties. The area covered by red and orange is the entropy of $X$, $H(X)$, whilst the area covered by yellow and orange represents the entropy of $Y$, $H(Y)$. The red area is the conditional entropy of X given by Y, $H(X|Y)$ whereas the yellow area is the conditional entropy of Y given by X, $H(Y|X)$. The area contained by both circles is the joint entropy $H(X,Y)$ and the orange area is the mutual information between $X$ and $Y$, $I(X;Y)$.

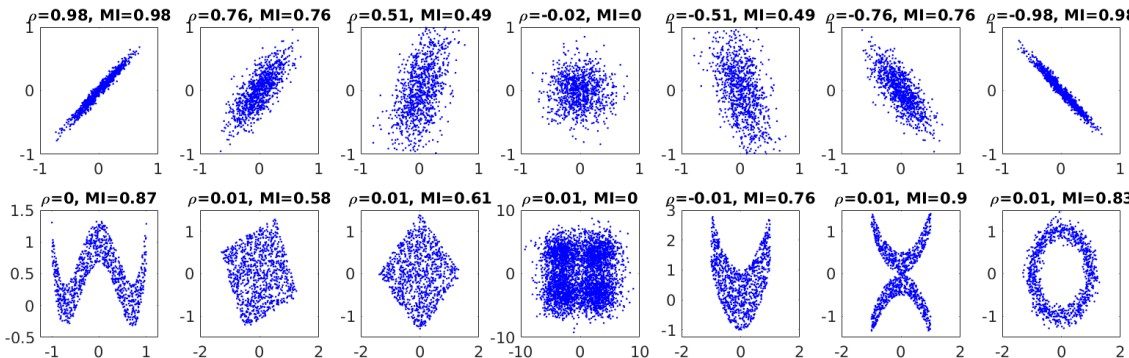

**Figure 3.** Mutual Information (MI) vs Pearson correlation ($\rho$) tested on linear (top) and non-linear (bottom) data sets. It can be seen that both methods are able to estimate the correlation for linear case. On the other hand, Pearson correlation fails in estimating the correlation for the non-linear data sets, whereas MI is capable in estimating the existence of correlation even in these cases. The so-called nearest neighbor implementation of mutual information method is used here (see text).



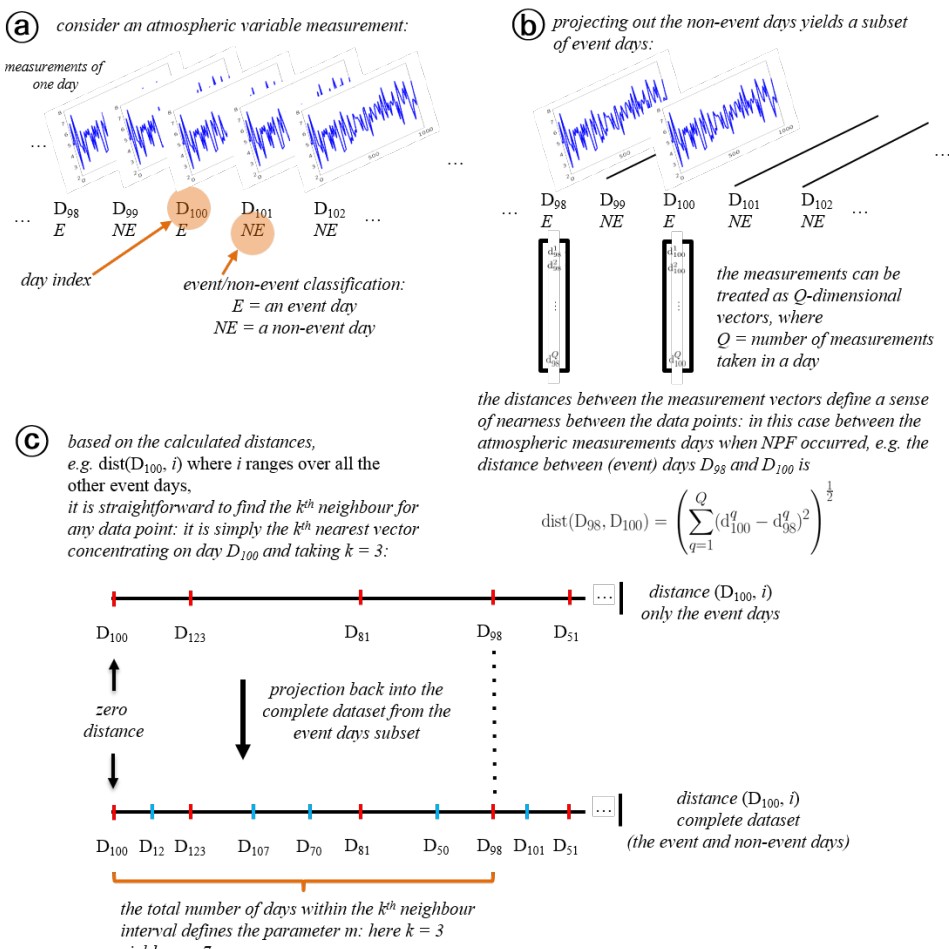

**Figure 4.** An illustration of the computing process for Mutual Information estimator based on nearest neighbor method.

0




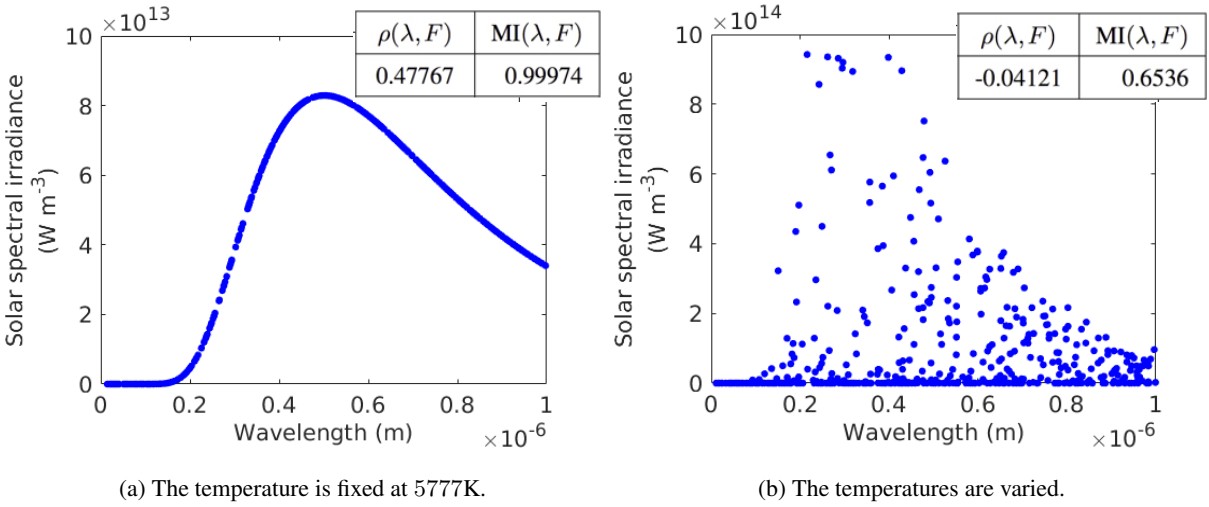

(a) The temperature is fixed at 5777K.  (b) The temperatures are varied.

**Figure 5.** The relationship between solar spectral irradiance and wavelength.

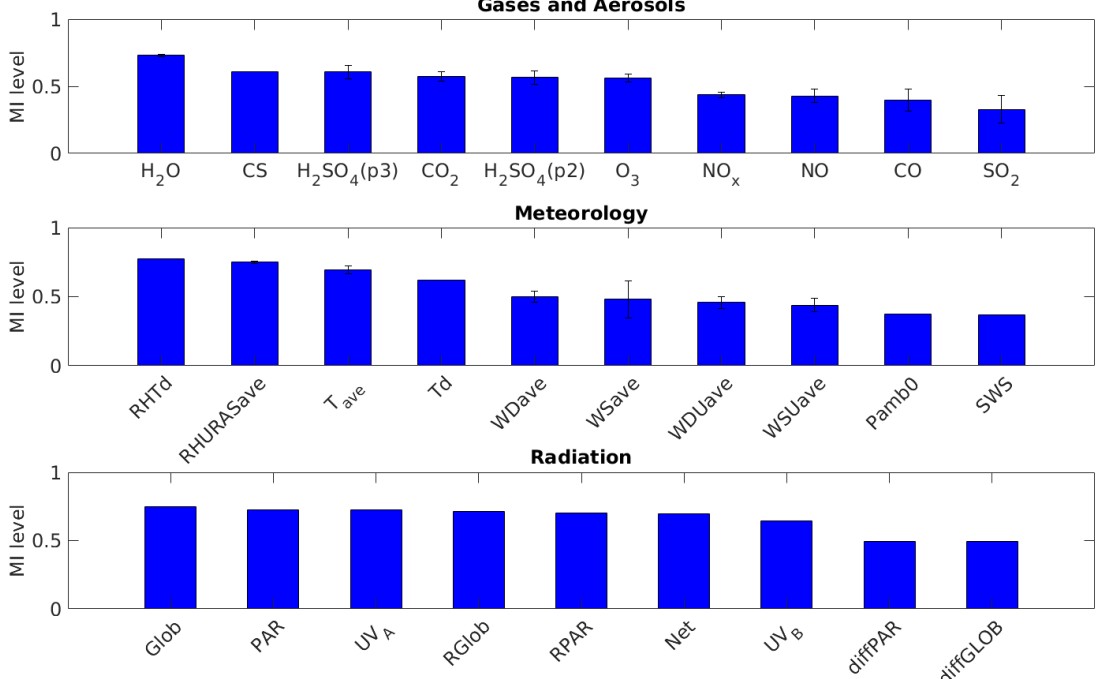

**Figure 6.** MI correlation level for a variety of atmospheric variables: gases concentration and aerosols (top), meteorology (middle) and radiation (bottom). It can be seen that water concentration ($H_2O$), condensation sink (CS), sulfuric acid ($H_2SO_4$), relative humidity ($RHT_d$), average temperature ($T_{ave}$) and global radiation (Glob) are among the atmospheric variables that have strong correlation to NPF.





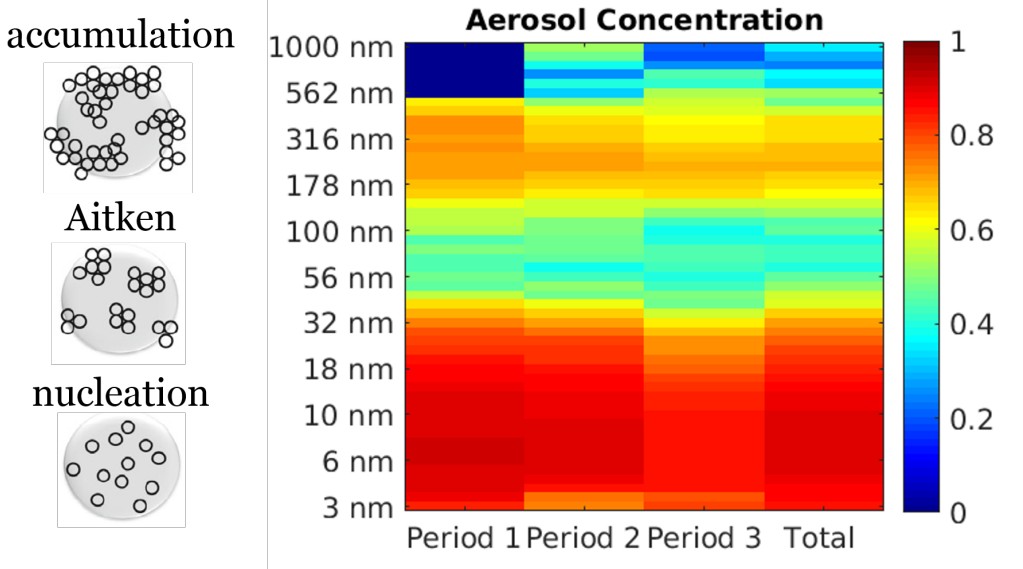

**Figure 7.** The correlation levels are obtained through MI method for the particle number size distribution at SMEAR II station in Hyytiälä. The colour shows the level of correlation. The first three columns represent three periods between years 1996-2014 where each period consists of the correlation level for 6 years. The last column indicates the total correlation level for 18 years.




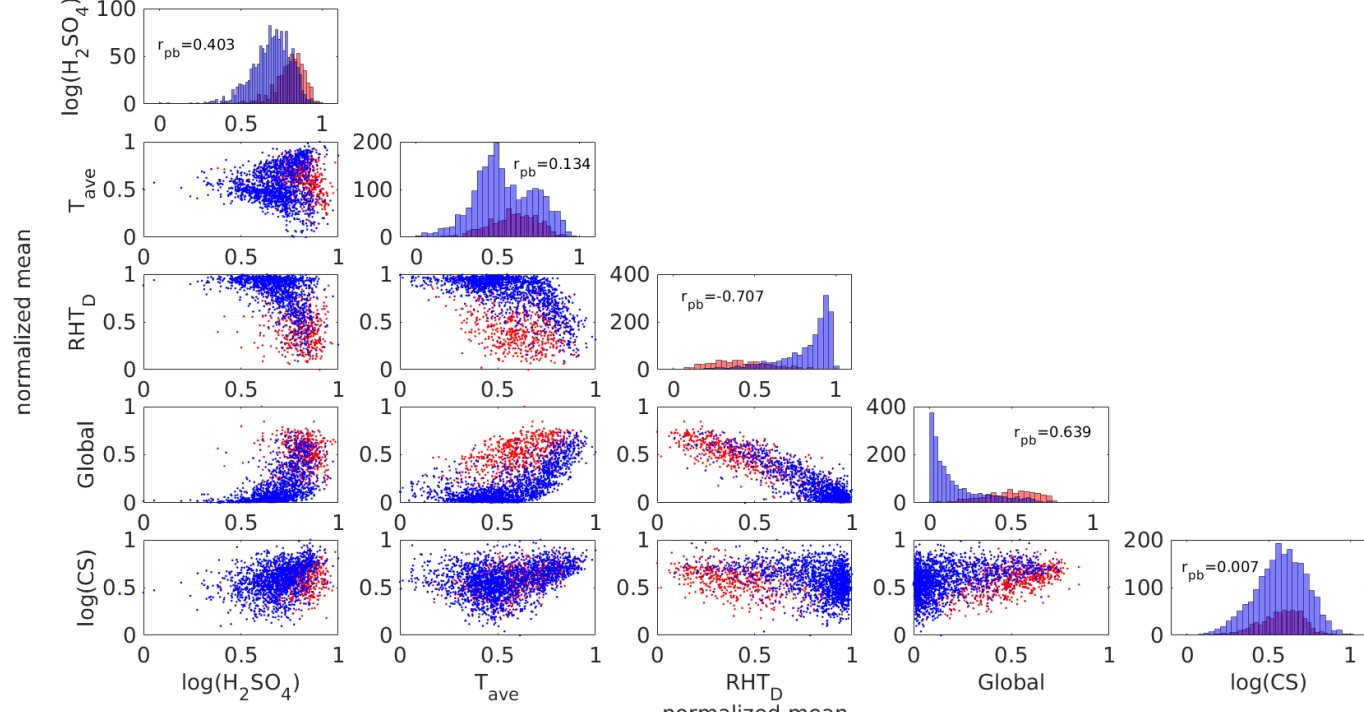

**Figure 8.** The scatter matrix plot between five selected atmospheric variables and NPF. The red and blue dots represent event and non-event days, respectively.



**Table**

| Atmospheric variables | Symbols |
|---|---|
| **Gas concentrations** | |
| nitrogen monoxide | NO |
| nitrogen oxides | $NO_x$ |
| ozone | $O_3$ |
| sulfur dioxide | $SO_2$ |
| water | $H_2O$ |
| carbon dioxide | $CO_2$ |
| carbon monoxide | CO |
| **Meteorology** | |
| Rain indicator | SWS |
| Dew point temperature at 16 m height | Td |
| Atmospheric pressure at ground level (180 m above sea level) | Pamb0 |
| Temperature | $T_{ave}$ |
| Wind speed | $WS_{ave}$ |
| Wind speed (sonic) | $WSU_{ave}$ |
| Wind direction | $WD_{ave}$ |
| Wind direction (sonic) | $WDU_{ave}$ |
| Relative humidity | $RHURAS_{ave}$ |
| Relative humidity at 16 m height | RHTd |
| **Radiation** | |
| Ultraviolet A | $UV_A$ |
| Ultraviolet B | $UV_B$ |
| Diffuse PAR | diffPAR |
| Diffuse shortwave radiation | diffGLOB |
| Net radiation | Net |
| Reflected global radiation | RGlob |
| Global radiation | Glob |
| Reflected PAR | RPAR |
| PAR, total | PAR |
| **Particles Concentration** | |

...





| | |
|---|---|
| Aerosol particle number size distribution (3 - 1000 nm) | 3nm - 1000nm |

Table 1: The name of used atmospheric variables, symbols displayed in the results and methods/measurement devices used in this study.