# Peer review of "Exploring nonlinear associations between atmospheric new-particle formation and ambient variables: a mutual information approach"

_Atmospheric Chemistry and Physics, 2018_

## Referee Comment (RC1) · Anonymous Referee #1 · 15 May 2018

In this study, an information theory approach has been used to analyze nonlinear correlations between different atmospheric variables (meteorological, aerosol, gas and radiation) and new particle formation (NPF) in Hyytiälä measurement station in Finland. The correlations have been evaluated by calculating the mutual information (MI) between different variables and event/non-event days using a nearest neighbor method developed earlier for other applications. The authors suggested the MI method could be widely used to evaluate correlations between different variables and NPF as well as other phenomena in the atmospheric science.

To my knowledge, this is the first study when an information theory approach (i.e. MI)

[Figure]

have been used in NPF analysis. Briefly, the results obtained show that NPF correlates with sulphuric acid and water concentrations, ultraviolet radiation, condensation sink and temperature. In the previous studies, those variables have also been associated to NPF. The MI seems to be a simple and effective method to analyses nonlinear correlations between ambient variables and NPF in large datasets without any supervision or prior knowledge of NPF mechanisms. Furthermore, it seems to be computationally relatively light and have only one free model parameter. Therefore, the method is a suitable tool to analyze large and complicated datasets in atmospheric science and other "Big data" applications. Especially, it can be used for screening suitable variables for more detailed analysis.

The manuscript (MS) is quite well organized and written. The content on the MS is in the scope of journal, although, e.g., Atmospheric Measurement Techniques could be a more relevant journal due to a technical aspect of the MS. Overall, the MS is suitable for publication in this journal because it introduces a new approach for atmospheric data analysis, especially for analysis of NPF involving complicated and nonlinear processes. However, some comments, suggestions and technical corrections should be considered and discussed before publication.

Specific comments:

Tittle

In title, "an information theoretic approach" is mentioned. I feel that "mutual information approach" or similar would be more descriptive.

Abstract (also results and discussion)

It is mentioned, "The applied mutual information method finds that the formation events correlate with sulfuric acid concentration..." Is there a specific level for MI (certain value) that indicates a correlation between different variables? Alternatively, are those variables the most correlating factors because they have the highest MI values? In

general, when you can say that some variables correlate or not when using the MI approach. Please discuss in MS text.

**1. Introduction**

In the introduction, the authors are citing only Hyvönen at al. (2005) for conducting similar data mining on parameters affecting NPF. It should be noted that Mikkonen et al. (2006, 2011) have used similar approachs, discriminant analysis and multivariate non-linear mixed effects model, to analyze key factors contributing to the NPF and growth of formed particles, respectively. They showed that in more polluted environments, like San Pietro Capofiume, Melpitz and Hohenpeissenberg, the parameters found in Hyytiälä were not sufficient to predict NPF. Especially, when Hyvönen et al. (2005) did not found global radiation to be important variable for NPF, in Mikkonen et al. (2006, 2011) papers it was the most important variable. Other significant parameters related to NPF were RH, O3, SO2, NO2 and temperature, some of these found relevant also in this study. I think that those studies should also be considered in the introduction and later in discussion together with the study by Hyvönen et al. (2005).

Please, also summarize very briefly other studies in the field of aerosol/atmospheric science that have used information theory approaches or a MI method. Is there any specific area in which those methods have been used frequently (e.g. remote sensing)? After quick search, I found some previous studies: Preining (1971), Li et al. (2009, 2012), and Brunsell and Young (2008) but probably there are much more published studies.

**2.2 Measured variables**

Please, check the size range and measurement height of the particle number size distribution measurements. For instance, Nieminen et al. (2014) mentioned that size range was 3-500 nm until Dec 2004 and Dal Maso et al. (2005) that sampling height was 2 m above ground.

2.3 Derived variables

Condensation sink has been calculated from particle size distributions, which size ranges were not same for all measurement. Has this any effect on the results?

Proxy concentration of sulfuric acid has been calculated from other variables. Does this have any effect on the results (MI values, relative correlations)?

For simplicity, undefined days have been removed before MI calculations. What would be an effect on the results if the undefined days were included in MI calculations? I think that MI can easily be used to evaluate several discrete variables.

3.1 Data pre-processing

You have normalized continuous variables to have zero mean and unit variance so that large numerical values are not too significant in analysis. In general, is this always needed if you use a Euclidean distance in the nearest neighbor method when calculating the MI values as described in Fig. 4?

In the analysis, you have eliminated nighttime data points in atmospheric variables. I suppose that after that you have not exactly same number of data points at exactly same time when calculating distances between variables (see Fig. 4b). Please, clarify in the MS how you have considered this problem and what is time resolution for measured variables (hour average/1 min instantaneous).

3.2 Information Theory: A brief introduction

I feel that Shannon, the pioneer in information theory, should be mentioned in the MS (Shannon, 1948). I suppose that his pioneer work is not well known for typical readers of this journal.

3.2.2 Mutual Information

In the Fig. 3, MI and Pearson correlation coefficient are shown a standard test set. Is there any reference for that data set or is it publicly available (line 19 page 7)?
Furthermore, a comparison of MI results to the Pearson correlation is a bit question-able as assumptions of the Pearson correlation are not valid for these data and, e.g., the Spearman correlation should be used instead. It might not change the results significantly but the comparison would be more valid.

3.3 Mutual Information implementation: nearest neighbor method

Please insert suitable references for that chapter. I think that is not generally known in the field of atmospheric science. The used nearest neighbor method have been described, e.g., in papers by Kraskov et al. (2004) and Ross (2014) as mentioned in a previous chapter. Kraskov et al. (2004) described two different algorithms and the first one seems to be used here. The notations and equations seems to be exactly same than in Ross (2014). Please, indicate preferences in more detail in this chapter.

Have you calculated MI values only for event days or both event and non-event days? Do the calculated MI levels present the key factors contributing to the NPF or the factors that best separate event and non-event days form each other (or vice versa)? If you have calculated MI values only for event days, what are key factors contributing to the non-event days. Please clarify this in the MS because it is now unclear for me. Practically, is the discrete variable x a set of event days or a set of event and non-event days in the calculations? Furthermore, if you include undefined days in MI calculations, how does this affect the results? Have you already done any calculations with event, non-event and undefined days? Those results would be very useful when a capability of MI method in NPF analysis is evaluated and thus should be discussed in the MS.

Why have bivariate correlations only been inspected? During NPF, multiple different phenomena occur simultaneously and thus the analysis should be multivariate. Can multivariate analysis be conducted using the information theory approach?

Please indicate in MS text, which distance (Euclidean distance, I suppose) and k-value (3?) you have used in the calculations. Furthermore, indicate how you have practically calculated MI values (using Matlab/Python/etc. programs made by authors, commercial programs, programs distributed by Ross (2014) or otherwise). Finally, the publisher encourages authors to deposit software, algorithms, and model code on suitable repositories/archives whenever possible (see the journal Data policy).

3.4 Mutual information: a simulation case study

Is this chapter needed? Is this simulation case relevant with the NPF analysis? In Fig. 3, you have already shown capability of the MI method to find non-linear correlations and this is, I think, only one example more and therefore you can remove it.

Have you used same program in this or Fig. 3 cases than in NFP calculations? For me, this and Fig. 3 cases look like continuous variables vs. continuous variables cases whereas NPF calculations are discrete variables vs. continuous variables cases.

I think that tests with simulated event/non-event data would be more relevant than solar spectrum data with very large temperature variation. Have you done any studies with simulated event/non-event data?

4 Results

A better title of this chapter is Results and Discussion because the chapter also includes discussion of the results (not only results, see classical IMRaD structure).

4.1 Correlation analysis between atmospheric variables and NPF

Is it possible find using the MI method whether the correlation is positive or negative (i.e., is lower or higher value more favorable) in relevant situations?

You mentioned that the temperature is associated with many atmospheric variables. I think that chemical reactions that produce condensable species depend on temperature so it could be also mentioned.

You stated that wind direction have little correlation with NPF and discussed that small correlation persists due to pollution from the westsouth - west (station building and city of Tampere). How about a local sawmill and a power plant in Korkeakoski, located ca.

6 km southeast of Hyytiälä (see e.g., Liao et al. 2011; Williams et al., 2011; Lopez-Hilfiker, 2014). Could the sawmill and the power plant have any influence on NPF and can you see this in MI values?

5 Conclusions

You stated: "The method also contains only one free parameter (the number of nearest neighbours k) and its value does not affect the results significantly". Have you tested several k-values? If this is a generally known fact, please add a suitable reference.

Can MI method use to analyze long-term changes in NPF (e.g. due to climate change)?

You discussed about automatic event classification algorithms. Please note a recent paper by Joutsensaari at al. (2018).

Figure 5.

Please indicate in a caption what does sigma and MI mean.

Figure 6.

"MI correlation level for a variety of atmospheric variables": Should NPF be mentioned in caption (MI correlation levels between NPF and a variety ...). Also, indicate in caption that notations are shown in Table 1.

Figure 7.

Does blue color (MI=0) in large particles in Period 1 indicate that there is no data or no correlation? Please clarify this.

Figure 8.

Please indicate in caption what r_pb means.

Technical corrections:

Page 12, line 2: ...2017) . As... => ...2017). As...

Page 12, line 13: (e.g. . . .) => (i.e., . . .)

References:

Brunsell, N. A., and Young, C. B.: Land surface response to precipitation events using MODIS and NEXRAD data, Int. J. Remote Sens., 29, 1965-1982, 10.1080/01431160701373747, 2008.

Dal Maso, M., Kulmala, M., Riipinen, I., Wagner, R., Hussein, T., Aalto, P. P., and Lehtinen, K. E. J.: Formation and growth of fresh atmospheric aerosols: Eight years of aerosol size distribution data from SMEAR II, Hyytiälä, Finland, Boreal Environ. Res., 10, 323-336, 2005.

Hyvönen, S., Junninen, H., Laakso, L., Dal Maso, M., Grönholm, T., Bonn, B., Keronen, P., Aalto, P., Hiltunen, V., Pohja, T., Launiainen, S., Hari, P., Mannila, H., and Kulmala, M.: A look at aerosol formation using data mining techniques, Atmos. Chem. Phys., 5, 3345-3356, https://doi.org/10.5194/acp-5-3345-2005, 2005.

Joutsensaari, J., Ozon, M., Nieminen, T., Mikkonen, S., Lähivaara, T., Decesari, S., Facchini, M. C., Laaksonen, A., and Lehtinen, K. E. J.: Identification of new particle formation events with deep learning, Atmos. Chem. Phys. Discuss., https://doi.org/10.5194/acp-2017-1189, in review, 2018.

Kraskov, A., Stögbauer, H., and Grassberger, P.: Estimating mutual information, Physical review E, 69, 066 138, 2004.

Li, Y., Xue, Y., Guang, J., Wang, Y., and Mei, L.: A retrieval algorithm for aerosol optical depth from MODIS multi-spatial scale data based on mutual information, 2009 IEEE International Geoscience and Remote Sensing Symposium, V-489-V-492, 2009.

Li, Y. J., Xue, Y., He, X. W., and Guang, J.: High-resolution aerosol remote sensing retrieval over urban areas by synergetic use of HJ-1 CCD and MODIS data, Atmos. Environ., 46, 173-180, 10.1016/j.atmosenv.2011.10.002, 2012.

[Figure]

Liao L., Dal Maso M., Taipale R., Rinne J., Ehn M., Junninen H., Äijälä M., Nieminen T., Alekseychik P., Hulkkonen M., Worsnop D.R., Kerminen V.-M. & Kulmala M. 2011. Monoterpene pollution episodes in a forest environment: indication of anthropogenic origin and association with aerosol particles. Boreal Env. Res. 16: 288–303.

Lopez-Hilfiker, F. D., Mohr, C., Ehn, M., Rubach, F., Kleist, E., Wildt, J., Mentel, Th. F., Lutz, A., Hallquist, M., Worsnop, D., and Thornton, J. A.: A novel method for online analysis of gas and particle composition: description and evaluation of a Filter Inlet for Gases and AEROsols (FIGAERO), Atmos. Meas. Tech., 7, 983-1001, https://doi.org/10.5194/amt-7-983-2014, 2014.

Mikkonen, S., Lehtinen, K. E. J., Hamed, A., Joutsensaari, J., Facchini, M. C., and Laaksonen, A.: Using discriminant analysis as a nucleation event classification method, Atmos. Chem. Phys., 6, 5549-5557, https://doi.org/10.5194/acp-6-5549-2006, 2006.

Mikkonen, S., Korhonen, H., Romakkaniemi, S., Smith, J. N., Joutsensaari, J., Lehtinen, K. E. J., Hamed, A., Breider, T. J., Birmili, W., Spindler, G., Plass-Duelmer, C., Facchini, M. C., and Laaksonen, A.: Meteorological and trace gas factors affecting the number concentration of atmospheric Aitken (Dp = 50 nm) particles in the continental boundary layer: parameterization using a multivariate mixed effects model, Geosci. Model Dev., 4, 1-13, https://doi.org/10.5194/gmd-4-1-2011, 2011.

Nieminen, T., Asmi, A., Dal Maso, M., Aalto, P. P., Keronen, P., Petaja, T., Kulmala, M., and Kerminen, V. M.: Trends in atmospheric new-particle formation: 16 years of observations in a boreal-forest environment, Boreal Environ. Res., 19, 191-214, 2014.

Preining, O.: Information theory applied to the acquisition of size distributions, J. Aerosol Sci., 3, 289-296, https://doi.org/10.1016/0021-8502(72)90050-X, 1972.

Ross, B. C.: Mutual information between discrete and continuous data sets, PloS one, 9, e87 357, 2014.

Shannon, C.E. (1948), "A Mathematical Theory of Communication", Bell System Technical Journal, 27, pp. 379–423 & 623–656, July & October, 1948

Williams, J., at al.: an overview of meteorological and chemical influences, Atmos. Chem. Phys., 11, 10599-10618, https://doi.org/10.5194/acp-11-10599-2011, 2011.

---

## Referee Comment (RC2) · Anonymous Referee #2 · 17 May 2018

This study used the information theoretic approach (i.e. MI method) to explore nonlinear associations between atmospheric new particle formation (NPF) and ambient variables. The study demonstrates that the same results can be obtained by MI method which operates without supervision and physical insight. The authors suggest that the method is suitable to be implemented widely in the atmospheric field. The manuscript is well organized and written. As it's the first time an information theoretic approach used in NPF study, in my opinion, this manuscript is suitable for publication in ACP provided that the following comments are addressed.

General comments:

[Figure]

1. I recommend that some modifications need to be done for the introduction. The introduction has too many paragraphs. The fourth, fifth and sixth paragraph can be merged. In those paragraphs, the authors only cite Hyvonen et al., (2005) paper but take lots of sentences to describe their methods. I recommend that some other references (or methods) should be cited here and it's better to use only 1 or 2 sentences to summarize their methods. Moreover, comparing with other methods, in the introduction the authors need to explain why the information theoretic approach is better or more suitable method to analyze the atmospheric data related to NPF.

2. As the authors introduce a new method to analyze the long-term atmospheric data, in the MS text the advantages and disadvantages of this method compared to common methods should be discussed in detail. An additional section and figure would be better for this discussion.

Specific comments:

Page 4, Line 26: You don't need to mention Weber et al.'s proxy if they are not used in your paper.

Page 5, Line 16 & 17: The instructions about figure are not needed here.

Page 10, Line 14: Is the water concentration similar with the relative humidity? You can give some hypothesis based on chamber studies from references.

Page10, Line 27 & 28 & 29: I would say that the correlation with O3 is also related to the formation of OH and H2SO4.

Figure 1 'Hyytiala station' need to be changed into 'SMEAR II station'

Figure 7 The plots of nucleation, Aitken and accumulation in the left panel are not needed in this kind of figure. You can define those in the MS text. Please add the labels for y-axis and colorbar.

---

## Author Comment (AC2) · 27 Jul 2018

**Point-to-point response to referee 2:**

We thank the reviewers for their encouraging and positive comments. The original comments (requiring a response) are shown in boldface. Our responses will be intercalated and the final manuscript will be revised accordingly.

**Reviewer comments:**

**1 I recommend that some modifications need to be done for the introduction. The introduction has too many paragraphs. The fourth, fifth and sixth paragraph can**

[Figure]

**be merged. In those paragraphs, the authors only cite Hyvonen et al., (2005) paper but take lots of sentences to describe their methods. I recommend that some other references (or methods) should be cited here and it's better to use only 1 or 2 sentences to summarize their methods. Moreover, comparing with other methods, in the introduction the authors need to explain why the information theoretic approach is better or more suitable method to analyze the atmospheric data related to NPF.**

Thanks for your suggestion, we will merge and simplify the paragraphs. As you suggested, we will also add extra explanation related to that, including Mikkonen et al. (2006, 2011), whom have used discriminant analysis and multivariate non- linear mixed effects model, to analyse key factors contributing to the NPF and growth of formed particles, respectively. In addition to that, we already mentioned briefly the drawbacks of previous methods, where we said that MI will be used to overcome those issues.

**2 As the authors introduce a new method to analyze the long-term atmospheric data, in the MS text the advantages and disadvantages of this method compared to common methods should be discussed in detail. An additional section and figure would be better for this discussion.**

The common practice for finding correlation between atmospheric variables is through linear correlation, analysing it via scatter plot and histogram. We have discussed how MI can be advantageous in dealing with long-term atmospheric data in section 4.2. A result via scatter plot and histogram that contains the most important atmospheric variables in atmospheric process is also shown. In that case, we demonstrate that although the common methods are typically efficient in finding correlation, but through that case study, there are few cases where the common method may not always be suitable.

**Specific comments:**

**Page 4, Line 26: You don't need to mention Weber et al.'s proxy if they are not used in your paper.**

We will remove this as suggested.

**Page 5, Line 16 17: The instructions about figure are not needed here.**
Yes, we explained this in the Figure 7's caption already. We will remove the redundant entry as suggested.

**Page 10, Line 14: Is the water concentration similar with the relative humidity? You can give some hypothesis based on chamber studies from references.**
There is one notable difference between these two: water vapor concentration usually increases with the rise of ambient temperature (T) because warmer air can simply hold more water. So that quantity is much higher during summer than winter. Relative Humidity (RH) is scaled to the maximum water content of the air, so it does not care about seasonal variation of T. On the other hand, RH varies a lot diurnally (because T does and the water vapor concentration is more constant over a diurnal cycle).

**Page10, Line 27 28 29: I would say that the correlation with O3 is also related to the formation of OH and H2SO4.**
Thanks for your explanation. We will add and incorporate this into our result explanation.

**Figure 1 'Hyytiala station' need to be changed into 'SMEAR II station'**
We will change it.

**Figure 7 The plots of nucleation, Aitken and accumulation in the left panel are not needed in this kind of figure. You can define those in the MS text. Please add the labels for y-axis and colorbar.**
We will do this as you suggested.

---

## Author Response (AR1)

**Point-to-point response to referee #1:**

*We thank the reviewers for their encouraging and positive comments. The original comments (requiring a response) are shown in boldface. Our responses are intercalated, and the final manuscript have been revised accordingly. The changes are marked by the blue fonts.*

**Reviewer comments:**

**Specific comments:**

**Tittle**
**In title, "an information theoretic approach" is mentioned. I feel that "mutual information approach" or similar would be more descriptive.**
*We changed the title to be: "Exploring non-linear associations between atmospheric new-particle formation and ambient variables: a mutual information approach ".*

**Abstract (also results and discussion)**
**It is mentioned, "The applied mutual information method finds that the formation events correlate with sulfuric acid concentration. . ." Is there a specific level for MI (certain value) that indicates a correlation between different variables? Alternatively, are those variables the most correlating factors because they have the highest MI values? In general, when you can say that some variables correlate or not when using the MI approach. Please discuss in MS text.**

*In general, there is no specific level for MI or threshold that indicate a correlation between different variables which is also similar to Pearson correlation where this correlation value gives an only indication of the variables relationship. The value of MI depends on the distribution and the amount data. Unless mutual information gives very high value (very close to one) or a very low number (very close to zero), scientists need to make their own judgement about the variable correlation. In this case, similar variables are grouped based on their measurement types (traced gases, radiation, etc.), and their correlation level is ranked. The variables that have the highest mutual information level indicates that they are more favourable to NPF process compared to other variables. We intercalated the following explanation in the abstract and result section: In the abstract, instead of using word correlate with, we use words "are strongly linked to" and we have also intercalated the above statement in section 4.1.*

**1. Introduction**
**In the introduction, the authors are citing only Hyvönen at al. (2005) for conducting similar data mining on parameters affecting NPF. It should be noted that Mikkonen et al. (2006, 2011) have used similar approaches, discriminant analysis and multivariate non-linear mixed effects model, to analyze key factors contributing to the NPF and growth of formed particles, respectively. They showed that in more polluted environments, like San Pietro Capofiume, Melpitz and Hohenpeissenberg, the parameters found in Hyytiälä were not sufficient to predict NPF. Especially, when Hyvönen et al. (2005) did not found global radiation to be important variable for NPF, in Mikkonen et al. (2006, 2011) papers it was the most important variable. Other significant parameters related to NPF were RH, O3, SO2, NO2 and temperature, some of these found relevant also in this study. I think that those studies should also be considered in the introduction and later in discussion together with the study by Hyvönen et al. (2005).**

**Please, also summarize very briefly other studies in the field of aerosol/atmospheric science that have used information theory approaches or a MI method. Is there any specific area in which those methods have been used frequently (e.g. remote sensing)? After quick search, I found some previous studies: Preining (1971), Li et al. (2009, 2012), and Brunsell and Young (2008) but probably there are much more published studies.**

*We added more discussion about suggested publications into our introduction section. The introduction has been revised and improved. The additional discussion about the use of information theory in the field of atmospheric sciences has been added into section 3.2.*

**2.2 Measured variables**
**Please, check the size range and measurement height of the particle number size distribution measurements. For instance, Nieminen et al. (2014) mentioned that size range was 3-500 nm until Dec 2004 and Dal Maso et al. (2005) that sampling height was 2 m above ground.**

*It is true that the measured particle size ranges were between 3-500nm until Dec 2004, and after that, it was extended to cover the size range from 3 nm to 1000nm. The sampling height is 35 m since it was moved to the tower in 2015. Previously, it was 2 m above the ground. Since we used the data until 2014, we correct this to be 2 m above the ground. We have changed this in the manuscript as: "The measured aerosol particle number size distribution ranges were between 3 - 500 nm until December 2004, and after that, it has been extended to cover the size range from 3 nm to 1000 nm. The sampling height was at 2 m until 2015 when the instrument has been moved to the tower at 35 m."*

**2.3 Derived variables**
**Condensation sink has been calculated from particle size distributions, which size ranges were not same for all measurement. Has this any effect on the results? Proxy concentration of sulfuric acid has been calculated from other variables. Does this have any effect on the results (MI values, relative correlations)?**

*It is difficult to comment if different size ranges in CS calculation in the overall measurement is affected by the results. Nevertheless, we believe that this might impact only slightly the outcome because MI estimation is the average of MI for all data points. See equations (13) and (14). For sulfuric acid, we believe that other correlated variables used for calculating the proxy may have only a slight effect on the results. For example, the radiation is known to influence NPF, but in our calculation SO2 has the least correlation among traced gases to NPF. In this case, MI tries to compromise this and finding its mutual information for sulfuric acid.*

**For simplicity, undefined days have been removed before MI calculations. What would be an effect on the results if the undefined days were included in MI calculations? I think that MI can easily be used to evaluate several discrete variables.**

*What we mean "for simplicity" is that we removed undefined days to prevent extra bias added to our data because the undefined data cannot be unambiguously classified as either an event or non-event day. Undefined days may belong to event or non-event days if further investigation is made. We expect that MI result will not be reliable if we include this group since our focus is only to find the relationship between NPF and atmospheric variables. We have clarified this as: "In order to prevent bias in the data,  we do not* consider the undefined days because this group cannot be unambiguously classified as either an event or non-event day. Undefined days may belong to event or non-event days if further investigation is made. Therefore, those are removed from our database."

**3.1 Data pre-processing**
**You have normalized continuous variables to have zero mean and unit variance so that large numerical values are not too significant in analysis. In general, is this always needed if you use a Euclidean distance in the nearest neighbor method when calculating the MI values as described in Fig. 4?**
*Yes, you are right, we obtained the same results with and without normalization. We removed the normalization statement from there since it gives the same result.*

**In the analysis, you have eliminated nighttime data points in atmospheric variables. I suppose that after that you have not exactly same number of data points at exactly same time when calculating distances between variables (see Fig. 4b). Please, clarify in the MS how you have considered this problem and what is time resolution for measured variables (hour average/1 min instantaneous).**
*We strived to find a common resolution for the calculation. Since we perform bivariate analysis, between NPF and an atmospheric variable, the time resolution varies for every variable. If a variable is measured every10 minutes, it means we used 10 minute time resolution. We have added this statement in section 3.1*

**3.2 Information Theory: A brief introduction**
**I feel that Shannon, the pioneer in information theory, should be mentioned in the MS (Shannon, 1948). I suppose that his pioneer work is not well known for typical readers of this journal.**
*We mentioned Shannon's first work on this field in the manuscript.*

**3.2.2 Mutual Information**
**In the Fig. 3, MI and Pearson correlation coefficient are shown a standard test set. Is there any reference for that data set or is it publicly available (line 19 page 7)?**
*Fig.3 uses the standard test set data, that is publicly available. The data is made available under the Creative Commons CC0 1.0 Universal Public Domain Dedication. A sentence has been added there to clarify that the standard test set is publicly available.*

**Furthermore, a comparison of MI results to the Pearson correlation is a bit questionable as assumptions of the Pearson correlation are not valid for these data and, e.g., the Spearman correlation should be used instead. It might not change the results significantly but the comparison would be more valid.**
*The Spearman correlation gives similar result with Pearson correlation on this data. We have included Spearman correlation in Fig. 3.*

**3.3 Mutual Information implementation: nearest neighbor method**
**Please insert suitable references for that chapter. I think that is not generally known in the field of atmospheric science. The used nearest neighbor method have been described, e.g., in papers by Kraskov et al. (2004) and Ross (2014) as mentioned in a previous chapter. Kraskov et al. (2004) described two different algorithms and the first one seems to be used here. The notations and equations seems to be exactly same than in Ross (2014). Please, indicate preferences in more detail in this chapter.**
*Since this is a continuous - discrete case, the reference should be Ross (2014). We clarified this already in the text*

**Have you calculated MI values only for event days or both event and non-event days? Do the calculated MI levels present the key factors contributing to the NPF or the factors that best separate event and non-event days form**

each other (or vice versa)? If you have calculated MI values only for event days, what are key factors contributing to the non-event days. Please clarify this in the MS because it is now unclear for me. Practically, is the discrete variable x a set of event days or a set of event and non-event days in the calculations? Furthermore, if you include undefined days in MI calculations, how does this affect the results? Have you already done any calculations with event, non-event and undefined days? Those results would be very useful when a capability of MI method in NPF analysis is evaluated and thus should be discussed in the MS.

*We calculated MI values for both event and non-event days. The MI attempts to find the best factors/variables that can differentiate between event and non-event days, so those are the atmospheric variable influencing NPF. As described earlier, undefined days are excluded to prevent extra bias added to our data because this group cannot be unambiguously classified as either an event or non-event day. We have clarified this in section 4.1.*

**Why have bivariate correlations only been inspected? During NPF, multiple different phenomena occur simultaneously and thus the analysis should be multivariate. Can multivariate analysis be conducted using the information theory approach?**

*This method can perform only with bivariate case. To find interrelation correlations, we need to perform mutual information for every variable and make a plot matrix to analyse the impact of each variable.*

**Please indicate in MS text, which distance (Euclidean distance, I suppose) and k-value (3?) you have used in the calculations. Furthermore, indicate how you have practically calculated MI values (using Matlab/Python/etc. programs made by authors, commercial programs, programs distributed by Ross (2014) or otherwise). Finally, the publisher encourages authors to deposit software, algorithms, and model code on suitable repositories/archives whenever possible (see the journal Data policy).**

*We mentioned in the manuscript that we used Euclidean distance with k=3. The software is the extension of Ross program, where we added extra features, such as the Numata scaling factor, see page 8. The software may be published later in Python and/or Matlab on in the first author's Github and whenever possible in the ACP.*

3.4 Mutual information: a simulation case study
**Is this chapter needed? Is this simulation case relevant with the NPF analysis? In Fig. 3, you have already shown capability of the MI method to find non-linear correlations and this is, I think, only one example more and therefore you can remove it.**
**Have you used same program in this or Fig. 3 cases than in NFP calculations? For me, this and Fig. 3 cases look like continuous variables vs. continuous variables cases whereas NPF calculations are discrete variables vs. continuous variables cases.**
**I think that tests with simulated event/non-event data would be more relevant than solar spectrum data with very large temperature variation. Have you done any studies with simulated event/non-event data?**

*The nice thing about the second simulation study is that we know the underlying equation and shows the relationship between variables. This case study demonstrates how MI is able to estimate the relationship among input and output variables in the known model equations. The principle of continues-discrete MI method is also based on Kraskov (2004) and the simulation test was already done in Ross (2014). In other words,* Fig. 5, is a validation study whereas Fig. 3 is just based on data. *We do not perform any study yet with simulated event / non-event data as we do not have the model equations to simulate the event and non-event day. It can be an extension to the present work.*

**4 Results**
**A better title of this chapter is Results and Discussion because the chapter also includes discussion of the results (not only results, see classical IMRaD structure).**
*Thanks for the suggestion. We have changed the title of this section to be "Results and Discussion".*

**4.1 Correlation analysis between atmospheric variables and NPF**
**Is it possible find using the MI method whether the correlation is positive or negative (i.e., is lower or higher value more favourable) in relevant situations?**
*Unfortunately, this method does not detect negative/positive correlation. As stated in the conclusion, this method will not replace completely the standard method, instead this method should be used in the first place before performing a deeper data analysis method, such as through histogram and scatter plots. MI acts as a detecting mechanism and Causality testing at a later stage can be used to understand the direction of flow of information from one variable to another.*

**You mentioned that the temperature is associated with many atmospheric variables. I think that chemical reactions that produce condensable species depend on temperature so it could be also mentioned.**
*We have clarified this in section 4.1*

**You stated that wind direction have little correlation with NPF and discussed that small correlation persists due to pollution from the westsouth - west (station building and city of Tampere). How about a local sawmill and a power plant in Korkeakoski, located ca6 km southeast of Hyytiälä (see e.g., Liao et al. 2011; Williams et al., 2011; Lopez-Hilfiker, 2014). Could the sawmill and the power plant have any influence on NPF and can you see this in MI values?**
*Unfortunately, we may not be able to see this from MI values. As mentioned earlier, the function of MI is for early correlation detection (which we may miss via Pearson correlation due to the non-linearity in variable relationship). Extra analysis and plotting are still required to understand a particular phenomenon.*

**5 Conclusions**
**You stated: "The method also contains only one free parameter (the number of nearest neighbours k) and its value does not affect the results**

**significantly". Have you tested several k-values? If this is a generally known fact, please add a suitable reference.**
*Yes, the result does not change significantly for our case. This fact was also mentioned in Ross (2014). We added this reference there.*

**Can MI method use to analyze long-term changes in NPF (e.g. due to climate change)?**
*Probably yes, if we group NPF days into two categories based on their occurrence or frequency. Then, we compare between these groups and all atmospheric variables. Therefore, we may also learn what variables influence the increase of their occurrence, etc.*

**You discussed about automatic event classification algorithms. Please note a recent paper by Joutsensaari at al. (2018).**
*OK. noted. We discuss about the use of MI to include other factors in the ML classifier. The paper mentioned uses only aerosol particles images (banana plot).*

**Figure 5.**
**Please indicate in a caption what does sigma and MI mean.**
*Changed.*

**Figure 6.**
**"MI correlation level for a variety of atmospheric variables": Should NPF be mentioned in caption (MI correlation levels between NPF and a variety . . .). Also, indicate in caption that notations are shown in Table 1.**
*Good point. We did as suggested.*

**Figure 7.**
**Does blue color (MI=0) in large particles in Period 1 indicate that there is no data or no correlation? Please clarify this.**
*Yes, it was due to no data available. Good point, we clarified this by adding in the caption: "Note that the dark blue on period 1 for particles larger than 500 nm is due to unavailable data for that size ranges."*

**Figure 8.**
**Please indicate in caption what r_pb means.**
*OK, we added in the caption that "The notation r_pb is point-biserial correlation coefficient."*

**Point-to-point response to referee #2:**

*We thank the reviewers for their encouraging and positive comments. The original comments (requiring a response) are shown in boldface. Our responses are intercalated and the final manuscript have been revised accordingly. The changes are marked by the blue fonts.*

**Reviewer comments:**

**#1 I recommend that some modifications need to be done for the introduction. The introduction has too many paragraphs. The fourth, fifth and sixth paragraph can be merged. In those paragraphs, the authors only cite Hyvonen et al., (2005) paper but take lots of sentences to describe their methods. I recommend that some other references (or methods) should be cited here and it's better to use only 1 or 2 sentences to summarize their methods. Moreover, comparing with other methods, in the introduction the authors need to explain why the information theoretic approach is better or more suitable method to analyze the atmospheric data related to NPF.**

*Thanks for your suggestion, we merged and simplified the paragraphs. As you suggested, we also added extra explanation related to that, including Mikkonen et al. (2006, 2011), whom have used discriminant analysis and multivariate non- linear mixed effects model, to analyse key factors contributing to the NPF and growth of formed particles, respectively. In addition to that, we already mentioned briefly the drawbacks of previous methods, where we said that MI will be used to overcome those issues.*

**#2 As the authors introduce a new method to analyze the long-term atmospheric data, in the MS text the advantages and disadvantages of this method compared to common methods should be discussed in detail. An additional section and figure would be better for this discussion.**

*The common practice for finding correlation between atmospheric variables is through linear correlation, analysing it via scatter plot and histogram. We have discussed how MI can be advantageous in dealing with long-term atmospheric data in section 4.2. A result via scatter plot and histogram that contains the most important atmospheric variables in atmospheric process is also shown. In that case, we demonstrate that although the common methods are typically efficient in finding correlation, but through that case study, there are few cases where the common method may not always be suitable.*

**Specific comments:**

**Page 4, Line 26: You don't need to mention Weber et al.'s proxy if they are not used in your paper.**
*We removed this as suggested.*

**Page 5, Line 16 & 17: The instructions about figure are not needed here.**
*Yes, we explained this in the Figure 7's caption already. We removed the redundant entry as suggested.*

**Page 10, Line 14: Is the water concentration similar with the relative humidity? You can give some hypothesis based on chamber studies from references.**
*There is one notable difference between these two: water vapor concentration usually increases with the rise of ambient temperature (T) because warmer air can simply hold more water. So that quantity is much higher during summer than winter. Relative*

*Humidity (RH) is scaled to the maximum water content of the air, so it does not care about seasonal variation of T. On the other hand, RH varies a lot diurnally (because T does and the water vapor concentration is more constant over a diurnal cycle).*

**Page10, Line 27 & 28 & 29: I would say that the correlation with O3 is also related to the formation of OH and H2SO4.**
*Thanks for your explanation. This was added.*

**Figure 1 'Hyytiala station' need to be changed into 'SMEAR II station'**
*Noted.*

**Figure 7 The plots of nucleation, Aitken and accumulation in the left panel are not needed in this kind of figure. You can define those in the MS text. Please add the labels for y-axis and colorbar.**
*Noted.*

[revised manuscript text omitted]